# Machine Learning Inference of Gene Regulatory Networks in Developing *Mimulus* Seeds

**DOI:** 10.3390/plants13233297

**Published:** 2024-11-23

**Authors:** Albert Tucci, Miguel A. Flores-Vergara, Robert G. Franks

**Affiliations:** Department of Plant and Microbial Biology, North Carolina State University, Raleigh, NC 27695, USA; atucci@ncsu.edu (A.T.); maflores@ncsu.edu (M.A.F.-V.)

**Keywords:** endosperm, embryo, hybrid seed inviability, seed development, postzygotic reproductive barriers, *Mimulus erythranthe*, RTP-STAR, KBoost, RNA-seq, speciation

## Abstract

The angiosperm seed represents a critical evolutionary breakthrough that has been shown to propel the reproductive success and radiation of flowering plants. Seeds promote the rapid diversification of angiosperms by establishing postzygotic reproductive barriers, such as hybrid seed inviability. While prezygotic barriers to reproduction tend to be transient, postzygotic barriers are often permanent and therefore can play a pivotal role in facilitating speciation. This property of the angiosperm seed is exemplified in the *Mimulus* genus. In order to further the understanding of the gene regulatory mechanisms important in the *Mimulus* seed, we performed gene regulatory network (GRN) inference analysis by using time-series RNA-seq data from developing hybrid seeds from a viable cross between *Mimulus guttatus* and *Mimulus pardalis*. GRN inference has the capacity to identify active regulatory mechanisms in a sample and highlight genes of potential biological importance. In our case, GRN inference also provided the opportunity to uncover active regulatory relationships and generate a reference set of putative gene regulations. We deployed two GRN inference algorithms—RTP-STAR and KBoost—on three different subsets of our transcriptomic dataset. While the two algorithms yielded GRNs with different regulations and topologies when working with the same data subset, there was still significant overlap in the specific gene regulations they inferred, and they both identified potential novel regulatory mechanisms that warrant further investigation.

## 1. Introduction

The evolutionary innovation of the seed as a reproductive propagule in seed plants has provided plants substantial evolutionary advantages, such as enhanced desiccation resistance and improved dispersal capabilities [1]. Certain advantages of angiosperm seeds potentially spurred the rapid diversification of species that resulted in the extraordinary richness of extant flowering plant lineages observed today [2,3,4]. Angiosperm seeds are composed of three major compartments: the seed coat, the embryo, and the endosperm. The primary role of the endosperm is to supply nutrients to the developing embryo; furthermore, it influences the timing of seed germination through abscisic acid (ABA) signaling [5]. The capacity to delay germination until conditions are favorable confers a significant advantage to seed-bearing plants and likely facilitated their displacement of older plant lineages [2,3]. Moreover, since endosperm failure is a common cause of hybrid seed inviability [6,7], the breakdown of proper endosperm development has likely been responsible for the formation of many reproductive barriers between diverging angiosperm lineages. Taken together, this suggests that the endosperm plays a central role in the remarkable species richness of angiosperms [6,8]. Given an estimated emergence of flowering plants in the Early Cretaceous period, rapid radiation and speciation was required to achieve the diversity of ecology and morphology currently observed among angiosperms. Investigations into the transcriptional dynamics and regulatory mechanisms of endosperm development will further our understanding of the evolutionary history of angiosperms and may identify new causes of seed failure.

In an effort to uncover important gene regulatory mechanisms acting during endosperm development, we utilized bioinformatic approaches to create and analyze gene regulatory network representations of endosperm and seed development. Gene regulatory networks (GRNs) are characterized by a collection of “nodes”, which represent genes, and “edges”, which represent gene-to-gene regulatory relationships. In our case, these gene-to-gene regulatory relationships can represent a direct relationship—where a source gene codes for a transcription factor that binds to the promoter of a target gene—or can also represent an indirect relationship—where the expression of an upstream source gene impacts the expression of a target gene via another modality. These other modalities include but are not limited to a source gene impacting hormone signaling that affects the expression of the target gene and a source gene regulating another gene encoding a transcription factor that subsequently regulates the target gene. Therefore, edges in our inferred networks can represent genes whose expressions are linked in complex relationships that include molecular components that are not represented in the GRN.

GRN inference has the capacity to uncover previously unknown regulatory interactions and has been used to identify and highlight multiple key regulatory genes underlying interesting biological processes. In tomato, GRN inference was used to identify hub genes involved in the process of graft formation [9] and discover a gene underlying the control of grafting compatibility [10]. Though the identification of “hub” genes that are involved in the highest number of inferred regulations is an intuitive way to highlight genes in a GRN, because we know that certain regulation patterns, such as feed-forward loops, are common in biology, the identification of such repeated regulation patterns provides an opportunity to highlight genes within the GRN in a more informed manner [10,11,12]. These repeated regulation patterns are present as “network motifs” in our inferred GRNs.

Network motifs are specific patterns of nodes and edges (genes and regulatory relationships in our case) that can be repeated with the same configuration being formed by different nodes (genes). These network motifs can be used to describe the topology of the network or highlight specific connections contained within it. Enrichment analysis for these network motifs of inferred GRNs has been used to highlight key regulators of stem cell maintenance and division [13] and has also been used to elucidate genetic regulatory relationships controlling T-cell development [11].

To infer GRNs of endosperm development, we utilized an existing transcriptomic time-course dataset collected from developing seeds of the monkey flower, *Mimulus guttatus* [14]. To build upon the work by Flores-Vergara et al. [14], we used their previously collected time-course RNA-seq data to perform gene regulatory network (GRN) inference. *Mimulus* is an important genus within which to infer GRNs of endosperm development. Evidence suggests that mechanisms of inter-species hybrid seed inviability in *Mimulus* are, at least in part, endosperm-based [7,15,16], and endosperm-based mechanisms of hybrid seed inviability have been documented in a variety of angiosperms [6,17,18]. These endosperm-based hybrid seed inviability mechanisms underlie important reproductive barriers that reinforce or perhaps drive speciation events. Thus, an analysis of GRNs of endosperm development may uncover key network motifs or nodes associated with endosperm development that may be mis-regulated during aberrant hybrid seed development. Defining key GRN motifs may open up avenues to better study the evolution of such reproductive barriers and begin to explain the evolution and diversity of angiosperms, Darwin’s famous “abominable mystery” [19].

Our analysis allowed us to identify trade-offs between two different machine learning methods used for GRN inference. The first method we deployed was the RTP-STAR pipeline [12]. This pipeline utilizes the random forest algorithm GENIE3, which was the best performing algorithm in the DREAM4 *Multifactorial Network* and DREAM5 *Network Inference* challenges [20,21]. In our case, RTP-STAR builds upon the GENIE3 algorithm by leveraging the time-course aspect of our dataset to estimate the sign of inferred regulations [12]. The sign indicates the predicted effect of the regulation (i.e., whether the inferred regulation is increasing or decreasing expression of a downstream target). The second machine learning approach we deployed was KBoost, which incorporates a tandem kernel Principal Component Analysis regression and gradient boosting steps to infer gene regulatory networks from expression data [22]. While KBoost does not account for the time-course nature of our dataset, it did perform competitively relative to the GRN inference algorithms investigated in the DREAM4 and DREAM5 challenges while requiring substantially less computing time to work with large datasets. We observed an expected improvement in computational efficiency when using KBoost compared with RTP-STAR, and the differences between the two algorithms extended to the types of regulations inferred, as well as the overall topology of the resulting GRNs. This allowed us to generate a wider-ranging set of putative regulations that can be referenced for future investigations into *Mimulus* seed development. We also highlight specific inferred regulations and network motifs of interest involving auxin signaling.

## 2. Materials and Methods

We downloaded raw Illumina NextSeq 500 RNA-seq reads available from Flores-Vergara et al. [14] (GSE123424). The dataset included reads from RNA samples extracted from hybrid seeds from a compatible cross between two accessions of the *Mimulus guttatus* species complex. CSS4, a serpentine-adapted *M. guttatus* annual, served as the paternal plant, while SEC39, a facultatively selfing *Mimulus pardalis* annual, served as the maternal plant. Fertilized developing seeds were collected at 2, 4, 6, 7, and 8 DAP (days after pollination)—with unfertilized ovules collected for 0 DAP measurements.

We processed the RNA-seq reads by using a standard pipeline of adapter and low-quality read trimming with Trimmomatic (with the settings ILLUMINACLIP:TruSeq3-PE.fa:2:30:10; LEADING:3; TRAILING:3; SLIDINGWINDOW:4:15; MINLEN:36) and quality control assessment with FastQC to ensure no sequences were flagged as poor-quality ones after trimming [23,24]. The cleaned reads were then aligned to the TOL v5.0 *M. guttatus* reference genome by using STAR (with the settings outSAMattributes = NH HI AS nM jM jI, outFilterMismatchNoverReadLmax = 0.04, outFilterIntronMotifs = RemoveNoncanonical, chimSegmentMin = 40, outSAMmapqUnique = 60, twopassMode = Basic, and outSAMunmapped = Within KeepPairs) [25]. Where Flores-Vergara et al. [14] mapped these reads back to the *M. guttatus* v2.0 genome assembly, we chose to align reads to the newer TOL v5.0 *M. guttatus* reference genome because of the better annotation of that newer genome version. The quantification of fragments per kilobase per million read (FPKM) as a measurement of gene expression was then performed by using Cufflinks [26]. By using a default false discovery rate (FDR) threshold of 0.05, a total of 24,203 genes returned detectable levels of expression estimated through this step.

### 2.1. Endosperm Enrichment Filtering

Given the likely role of the endosperm in establishing hybrid seed inviability in *Mimulus* [7,15], we aimed to focus our initial GRN inference efforts on identifying potential regulatory relationships active in the endosperm. We generated a RNA-seq dataset derived from isolated hybrid endosperm tissue (data available at NCBI Sequence Read Archive under BioProject PRJNA1189279). Endosperm was hand-dissected from 9–10 DAP hybrid seeds derived from a reciprocal cross between two *M. guttatus* accessions—IM767 and CSS4—yielding seven biological replicates. The RNA-seq reads from this isolated endosperm dataset were processed similarly to the time-course dataset by using Trimmomatic and FastQC [23,24]. A differential expression analysis was then performed by contrasting the isolated endosperm dataset and RNA-seq reads from the 8 DAP whole-seed RNA-seq data from the Flores-Vergara et al. study.

In preparation for the differential expression analysis, reads from both datasets were aligned to the TOL v5.0 *M. guttatus* reference genome by using STAR (with the settings outSAMattributes = NH HI AS nM jM jI, outFilterMismatchNoverReadLmax = 0.04, outFilterIntronMotifs = RemoveNoncanonical, chimSegmentMin = 40, outSAMmapqUnique = 60, twopassMode = Basic, and outSAMunmapped = Within KeepPairs) [25,27]. Read count quantification was then performed by using featureCounts [28]. By utilizing the resulting gene expression estimates, differential expression analysis was executed by using edgeR [29]. Genes surpassing thresholds of a 1.5 $log_{2}$ fold change increase in estimated expression in the endosperm relative to the 8 DAP whole-seed data with an FDR-adjusted *p*-value under 0.01 were then categorized as “endosperm enriched”. This categorization of “endosperm enriched” served as an initial filtering step for the gene set used for the inference of GRN1 and GRN2 (list of gene IDs in this gene set can be found in Appendix A).

### 2.2. Modified Shannon Entropy Filtering

Filtering the set of expressed genes before GRN inference reduces the computational burden on the inference step and makes the prediction of spurious regulatory network relationships less likely. To achieve this filtering, we applied a Modified Shannon Entropy (MSE) approach as a means to both filter the entire gene set (used to infer GRN3 and GRN4) and serve as a second filter to the endosperm-enriched gene set (used to infer GRN1 and GRN2). For the gene set used for both GRN1 and GRN2, an initial filtering step was performed by excluding genes that did not get categorized as “endosperm enriched”. Entropy scores were calculated for each gene across the time course by using the method previously described in Kadota [30] and Thomas and Van den Broek [10]. Following the outlier detection scheme described by Thomas and Van den Broek [10], once each expression value for each gene was assigned an entropy score, any entropy score less than 30% was labeled as an outlier value. Then, any gene that recorded an outlier expression value in at least one time point across the time series was retained.

### 2.3. Transcription Factor Annotation

To select for potential transcription factors (TFs) in our gene set, we identified genes annotated with identifiers associated with TF activity or DNA-binding domains. These annotations included gene ontology (GO) terms, InterPro IDs, PFam IDs, and PANTHER IDs (the identifiers used can be found in Appendix A). The annotations for each gene were pulled from the accompanying annotation file for the Joint Genome Institute *M. guttatus* TOL v5.0 reference genome. The Joint Genome Institute reference database was then used to supplement this list of annotated TFs through a keyword search for “transcription factor” and “DNA binding”. A total of 6234 genes were identified as potential TFs, 4997 of which were found in our total expressed gene set and used to generate GRN5 and GRN6.

### 2.4. Gene Regulatory Network Inference

Each of the three gene sets (endosperm enriched + MSE filtering, whole gene set with MSE filtering, and transcription factors) was used for GRN inference using both RTP-STAR [12,31] and KBoost [22] (all networks are depicted in Appendix A). All GRN networks were inferred by using FPKM values calculated by Cufflinks [26] for each replicate in our time-course RNA-seq dataset. RTP-STAR was deployed by using the tuxnet suite [12], and the mean values of gene expression at each time point were used to assign the sign of inferred regulatory relationships (promoting, repressing, or just regulating). Five iterations of RTP-STAR were used to infer GRN1; however, due to the computational demands when working with the two larger gene sets, only one iteration was used to infer GRN3 and GRN5. Since there was no spatio-temporal clustering performed upstream of our analysis, the number of iterations performed should not have a significant impact on our analysis [12]. KBoost was deployed by using the KBoost R package available through Bioconductor [32] and did not incorporate the time-course element of the RNA-seq dataset. Since we were not updating based upon a previously validated dataset of genetic regulations in *Mimulus*, no prior weights were applied during GRN inference with KBoost. Regulator genes with the highest posterior probability value for each gene were determined by the default KBoost model and used as inferred GRN regulations for downstream analysis.

### 2.5. Network Motif Analysis

The detection of overrepresented motifs in all six of the inferred GRNs (GRN1, GRN2, GRN3, GRN4, GRN5, and GRN6) was performed by using QuateXelero [33]. For each network, QuateXelero generated 1000 random networks of the same size and calculated the number of times subgraph patterns (network motifs) occurred in each one. This allowed us to compare the frequency of these motifs in the GRN relative to their frequency in randomly generated networks with the same number of nodes and edges. For network motifs that were significantly overrepresented in the GRN, the instances of the subgraphs following the motif structure were identified in the GRN by using igraph [34].

## 3. Results

Time-series RNA-seq data from whole hybrid seeds of a *M. guttatus* (CSS4) × *M. pardalis* (SEC39) cross [14] were preprocessed and mapped to the TOL v5.0 *M. guttatus* reference genome (see Section 2). A total of 24,203 genes (hereafter the “expressed gene set”) were mapped and had expression quantified at detectable levels. This expressed gene set was further filtered to form three different gene sets, each of which was used to infer two GRNs—one inferred with RTP-STAR and one with KBoost.

Of the three gene sets used for GRN inference, the first included only genes that were identified as endosperm-enriched by using differential expression analysis and were subsequently filtered by using a Modified Shannon Entropy approach (this first gene set was used to infer GRN1 and GRN2). To generate the second gene set, a modified Shannon entropy filtering step was applied to the entire expressed gene set (this second gene set was used to infer GRN3 and GRN4). The third gene set (used to infer GRN5 and GRN6) consisted of all genes annotated as transcription factors (Figure 1 and Section 2). For each inferred GRN, we detected three-node network motifs that were over represented in the network and identified specific instances of these network motifs within each network to elucidate potential genes of interest for future analyses.

### 3.1. Endosperm-Enriched Genes

From the whole set of 24,203 expressed genes, 14% were categorized as endosperm-enriched through a differential expression analysis comparing “isolated endosperm” (endosperm hand-dissected from *Mimulus* seed) RNA-seq samples and “whole seed” (complete, intact *Mimulus* seed) RNA-seq samples (Figure 2). These 3456 genes were classified as endosperm-enriched by using a threshold of 1.5 $log_{2}$ (fold change) in expression and an FDR-adjusted *p*-value lower than 0.01. Among these endosperm-enriched genes were a number of genes homologous to *Arabidopsis thaliana* genes with previously identified roles in endosperm development: *HAIKU1* (AT2G35230) homolog MgTOL.E0874 (FDR-adjusted *p*-value = $4.55\times 10^{-4}$, $log_{2}$ (fold change) = 1.61), *WDR55* (AT2G34260) homolog MgTOL.B0405 (FDR-adjusted *p*-value = $7.72\times 10^{-4}$, $log_{2}$FC = 1.74), and *TITAN-LIKE* (AT4G24900) homolog MgTOL.G0604 (pFDR-adjusted = 2.90 $\times 10^{-6}$, $log_{2}$FC = 2.87) [35,36,37]. Similar to GRN1, prior to performing the network inference step, an MSE filtering step was applied to narrow the search space [10,12,38]. The final MSE-filtered, endosperm-enriched gene set used for GRN inference consisted of 317 genes and generated GRN1 and GRN2.

### 3.2. Network Motif Detection

Network motifs containing three nodes were detected and tested for over-representation by using QuateXelero by comparing the abundance of specific network motifs in the inferred GRN to 1000 random GRNs of the same size in terms of number of nodes and edges [33]. Network motifs were determined to be overrepresented (or enriched) in the inferred GRN if they had a Z-score over 3.0 according to permutation testing against 1000 randomly generated networks. Additionally, for each inferred GRN, we calculated a network motif score (NMS) for every gene present in at least one occurrence of an enriched network motif. These NMS values reflect the total number of network motif occurrences that a gene was present in for a given network (Appendix A contains an exhaustive breakdown of the network motifs identified in each GRN, and Appendix A provide a table for the genes with the highest network motif scores for GRN1-GRN6).

### 3.3. Difference Between Inference Algorithms

The identification of network motifs present in each of the inferred GRNs highlighted substantial differences in the topology of networks inferred by using RTP-STAR vs. networks inferred by using KBoost. While similar in the total number of edges inferred, KBoost networks were generally less interconnected, as there were fewer genes overall that had large numbers of inferred regulators (Appendix A). This is reflected in the substantially lower value of total network motifs identified in KBoost GRNs relative to RTP-STAR GRNs (Figure 3). This is likely due to the way in which edges were defined by using the KBoost model, where only regulators with among the highest posterior probabilities were assigned an edge to the putative target genes.

Both RTP-STAR and KBoost inferred similar regulations when using the same gene set, despite systematic differences affecting GRN topology (Figure 4). To determine the statistical significance of the degree of overlap detected when using RTP-STAR and KBoost, permutation testing was performed for each pair of inferred networks (GRN1 and GRN2, GRN3 and GRN4, and GRN5 and GRN6) by sampling two genes without replacement from the common gene set used for inference. This selection of random gene pairings was repeated until the number of these pairings matched the number of one of the inferred GRNs being compared. This collection of random gene pairings was then measured against the other GRN being compared to determine the degree of overlap. This process was repeated 10,000 times for each pair of inferred networks for permutation testing. This analysis indicated that for each of the three datasets analyzed, the regulatory edges inferred by RTP-STAR and KBoost overlapped to a greater extent than would be expected by random chance (*p*-value < 0.001). However, both RTP-STAR and KBoost displayed a proclivity to identify similar regulations regardless of the total composition of the gene set used for the analysis. For example, even though GRN2 and GRN4 were based on two different gene sets, these two KBoost-inferred networks had more regulatory interactions in common with each other than either had with any RTP-STAR network.

## 4. Discussion

Our analysis expands upon and provides further detail to the findings of Flores-Vergara et al. [14]. The clustering analysis performed by Flores-Vergara et al. identified potential key regulators of gene clusters associated with specific biological processes. Many of these highlighted potential regulators were also recovered in enriched network motifs in our analysis, including a *FUSCA3* signaling motif, as well as a secondary cell wall signaling motif involving both *SECONDARY CELL WALL-ASSOCIATED NAC DOMAIN2* and *KNAT7*. Our analysis supports these observations and extends the previously published work by inferring putative downstream targets of these regulators of interest and elucidating specific genes for future functional studies. Though, it should be noted that the putative regulations predicted in these GRNs may represent indirect relationships between genes.

### 4.1. Network Motifs of Interest in Endosperm-Enriched Networks GRN1 and GRN2

Between RTP-STAR and KBoost, a total of 1370 unique regulations were inferred by using the endosperm-enriched gene set. Many of these putative regulations could spur further investigation as potential active regulatory mechanisms in the developing *Mimulus* seed. However, one specific motif of interest identified in GRN1 involves a feedback loop between two genes that is regulated by a third mediator gene (depicted in Figure 3, motif 2), which allowed us to identify a putative regulatory relationship between MgTOL.F0024, a homolog of *FUSCA3* (AT3G26790) in *A. thaliana*, and MgTOL.E0523, a homolog of *ZINC FINGER PROTEIN 4* (AT1G66140) in *A. thaliana*. Both *FUS3* and *ZFP4* have been previously linked to processes in seed development and growth pathways. *FUS3* has been classified as an imprinted gene with paternally biased expression in the *A. thaliana* embryo [39], and *FUS3* is known to regulate the expression of seed storage proteins (SSPs) and to be necessary for proper endosperm development and maintenance of embryonic cell fate [40]. Importantly, *FUS3* also represses the biosynthesis of gibberellin (GA) [41,42]. Conversely, *ZFP4* has been identified as a downstream target of GA signaling through modulation via DELLA proteins [43]. Therefore, both MgTOL.F0024 and MgTOL.E0523 are homologous to genes linked to the GA signaling pathway in *A. thaliana* and are specifically homologous to genes lying on opposite ends of that cascade (*FUS3* acts upstream of GA biosynthesis, while *ZFP4* acts downstream of GA). MgTOL.F0024 and MgTOL.E0523 may be involved in GA-signaling- and regulation-related processes during endosperm or seed development in *Mimulus* species.

Notably, this motif was only identified in the RTP-STAR-inferred GRN1; however, by using the same gene set, both RTP-STAR and KBoost identified another putative developmental regulation involving MgTOL.N1182, the homolog of *ZHOUPI* (AT1G49770) in *A. thaliana*. *ZHOUPI* has been demonstrated to be necessary to embryonic development [44], and MgTOL.N1182 has qRT-PCR-validated enrichment in expression in the *Mimulus* endosperm (Flores-Vergara unpublished data). In both GRN1 and GRN2, MgTOL.N1182 had an inferred regulatory interaction with MgTOL.D0110—a homolog of AT1G02700, a GATA-like transcription factor in *A. thaliana*. This regulatory relationship has been validated as a primary regulation in *Arabidopsis* via AthaMap [45,46], and both *ZHOUPI* and GATA-like transcription factors are known downstream targets of *PHERES1* signaling in the developing *Arabidopsis* endosperm [47].

### 4.2. Network Motifs of Interest in MSE Networks GRN3 and GRN4

Broadening the search space to the entire gene set of 24,203 expressed genes allowed us to overcome many though not all of the thresholding issues that limited the regulatory relationships that we could infer in GRN1 or GRN2. *PEG2* (AT1G49290) in *A. thaliana* is a paternally expressed imprinted gene that has a demonstrated role in hybrid seed failure [48]. Though the connection is likely indirect, the expression of *PEG2* and its impact on hybrid seed failure seem to be linked to auxin production, as auxin-overproducing seeds exhibit elevated levels of *PEG2* expression in *A. thaliana* [47]. A similar relationship between auxin production and *PEG2* expression was inferred in both GRN3 and GRN4, as the *Mimulus* ortholog of *PEG2*, MgTOL.F1860, was linked in a reciprocal regulatory relationship to MgTOL.E1399, a homolog of *YUCCA2* (AT4G13260) in *A. thaliana*, an enzyme involved in the biosynthesis pathway of auxin [49,50].

### 4.3. Network Motifs of Interest in Transcription Factor Networks GRN5 and GRN6

By specifically looking for relationships among transcription factors in GRN5, we improved our likelihood of inferring direct regulatory relationships between genes. Among these regulations highlighted through the network motif enrichment of GRN5 was the repeated identification of known regulatory relationships involved in secondary cell wall synthesis. MgTOL.H1470 is homologous to *SECONDARY CELL WALL-ASSOCIATED NAC DOMAIN2* (*SND2*) in *A. thaliana* (AT4G28500). The role of *SND2* and related NAC domain proteins in activating secondary cell wall development is well documented [51,52]. One downstream target of these key regulators of secondary cell wall biosynthesis that has been repeatedly validated is *KNAT7* (AT1G62990) in *A. thaliana* [51,53]. The *KNAT7* homolog MgTOL.E0954 is present in multiple enriched network motifs as a downstream target. One such motif (matching the pattern of motif 4 in Figure 3) contains MgTOL.H1470, reciprocally regulating MgTOL.E0954 and *MYB43* (AT3G08500) homolog MgTOL.K0844 (node A), which has also been shown to be a downstream target of SND proteins in *A. thaliana* [51,54]. The inference of these potentially conserved regulations suggests that these regulatory mechanisms may be shared between *Arabidopsis* and *Mimulus*, even though the cellular characteristics of endosperm development (syncytial versus ab initio cellular) in *Arabidopsis* and *Mimulus* species are different.

Given the interval between the developmental time series (approximately two days), it is likely that the regulatory relationships predicted in these GRNs include both direct and indirect relationships—not every inferred regulation in our GRNs is reflective of a putative binding of a transcription factor to the promoter of the regulated target gene. However, the biological importance of indirect relationships may still be significant and useful in the process of defining modules of important functional network motifs. Future experimental validation of these predicted regulatory relationships will be important to confirm the role of these genes in endosperm and seed development in *Mimulus*.

## Figures and Tables

**Figure 1 plants-13-03297-f001:**
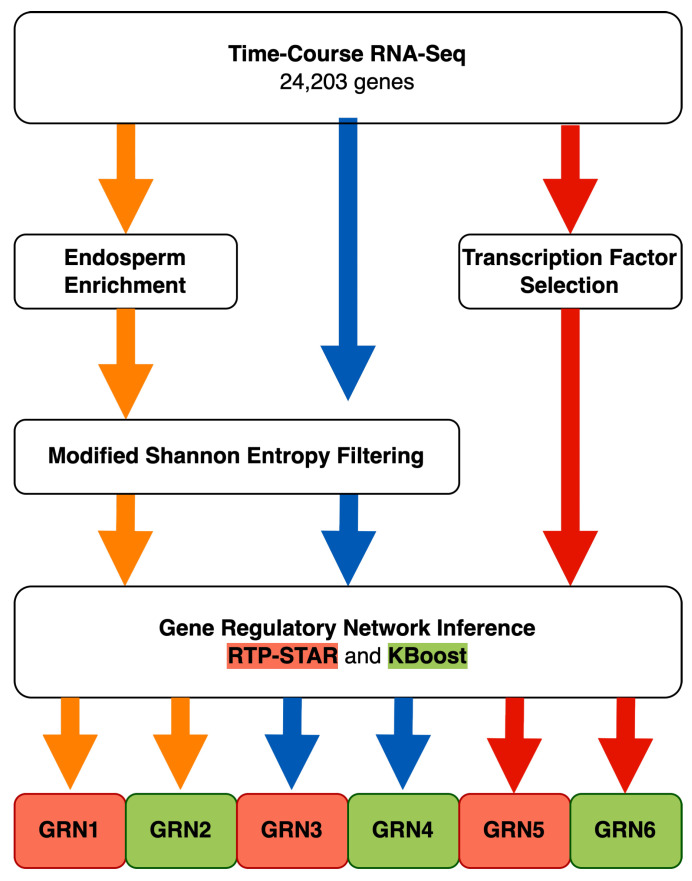
Workflow of data preparation for gene regulatory network (GRN) inference. Six separate GRNs were inferred by using three distinct sets of genes, all based upon time-course RNA-seq data of developing *Mimulus* seeds. The first gene set was restricted to “endosperm-enriched” genes before undergoing MSE filtering. The second gene set included all 24,203 genes with detected expression over the time course and filtered via MSE. The last gene set was a subset of predicted transcription factors and was the only gene set that did not undergo MSE filtering.

**Figure 2 plants-13-03297-f002:**
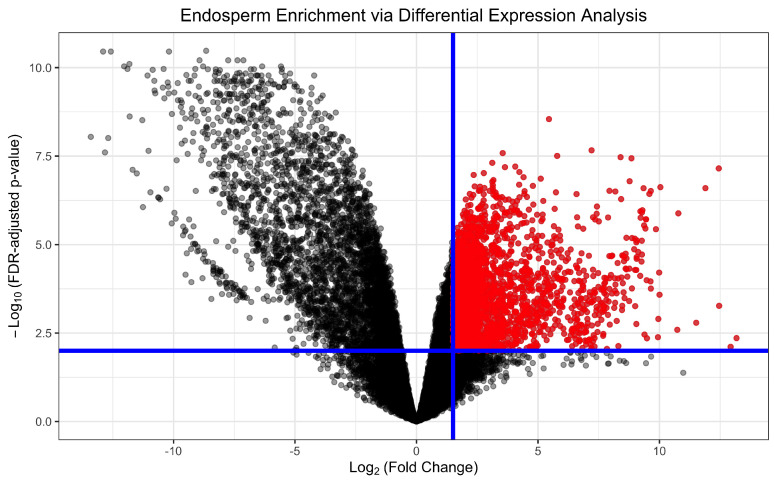
Identification of an endosperm-enrichment gene set through differential expression analysis of RNA-seq data from whole-seed samples at day 8 and endosperm isolated samples at day 8. The volcano plot is marked with blue lines indicating significance thresholds of FDR-adjusted *p*-value ≤0.01 and $log_{2}$ (fold change) > 1.5. Genes highlighted in red were labeled “endosperm enriched” and included in downstream analysis.

**Figure 3 plants-13-03297-f003:**
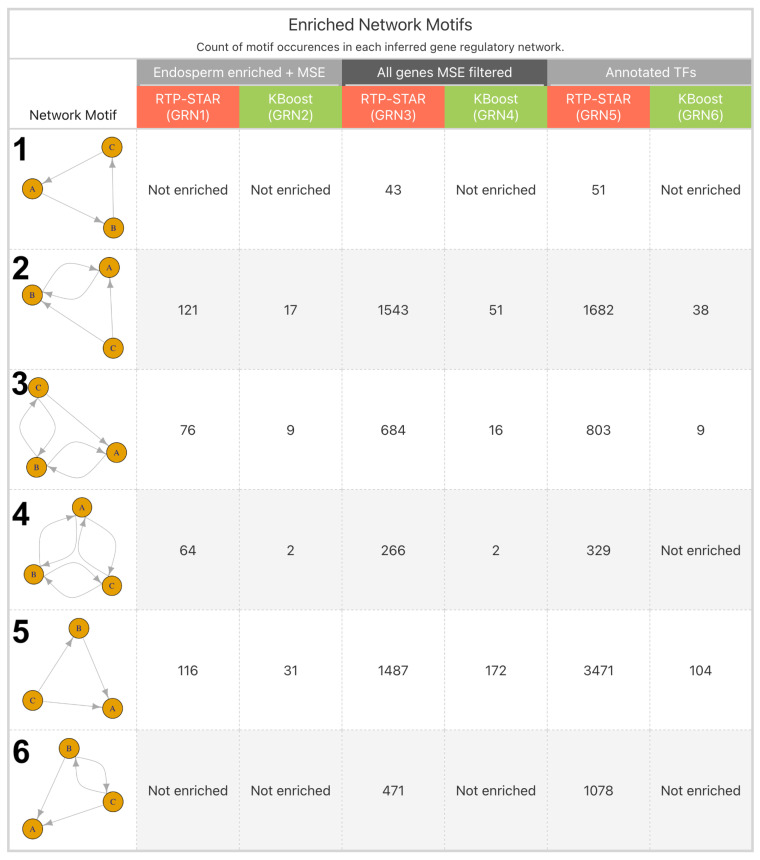
Breakdown of network motif enrichment analysis of each of the 6 inferred GRNs. Network motif enrichment was performed by using QuateXelero to identify network motifs in each GRN and subsequently identify which motifs were overrepresented in each GRN relative to random networks of the same size. All 6 GRNs were enriched for at least 3 of 6 distinct network motifs.

**Figure 4 plants-13-03297-f004:**
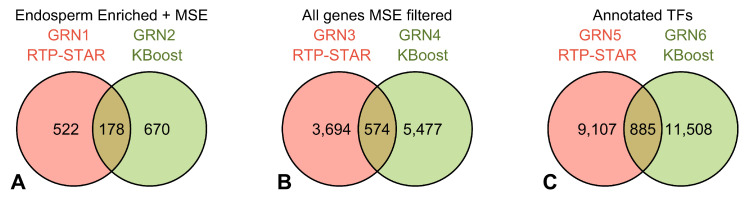
Overlap between networks when using the two separate machine learning approaches RTP-STAR and KBoost. (**A**) RTP-STAR inferred 700 putative regulations by using the endosperm-enrichedgene set for GRN1, and 178 (25.4%) of these regulations were also inferred by using KBoost for GRN2. (**B**) RTP-STAR inferred 4268 putative regulations by using the MSE filtered gene set for GRN3, 574 (13.4%) of which were also inferred by using KBoost for GRN4. (**C**) A total of 9992 putative regulations were inferred with RTP-STAR by using the annotated TF gene set for GRN5, 885 (8.9%) of which were also inferred by using KBoost on the same gene set for GRN6. For all three network comparisons, the amount of overlap between the RTP-STAR and KBoost networks was found to be statistically significant (*p*-value < 0.001 for each) through permutation testing.

## Data Availability

Sequencing data for this project can be obtained at GSE123424 and at the NCBI Sequence Read Archive under BioProject PRJNA1189279; the code used for this project is available at https://github.com/atucciNCSU/GRN_Mimulus_WS_timecourse (accessed on 18 November 2024).

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
