# Peer review of "Machine Learning Inference of Gene Regulatory Networks in Developing Mimulus Seeds"

_plants, 2024, doi:10.3390/plants13233297_

Round 1

Reviewer 1 Report

Comments and Suggestions for Authors

Title: “Machine Learning Inference of Gene Regulatory Networks in Developing Mimulus Seeds” correct to “Machine Learning Inference of Gene Regulatory Networks During Mimulus Seed Development”

a)    Abstract

i)               Clarity and Specificity:

  • “The angiosperm seed represents a critical evolutionary breakthrough”: It could be clearer to specify what this breakthrough entails, such as explaining how it facilitates reproductive success.
  • “likely helped propel the reproductive success and radiation of flowering plants”: The phrase "likely helped" introduces uncertainty. If there is strong evidence, it would be more assertive to say "has been shown to propel."

ii)             Sentence Structure:

  • “One way seeds facilitate the rapid diversification of angiosperms is through establishing postzygotic reproductive barriers in the form of hybrid seed inviability”: This sentence is somewhat convoluted. It could be simplified for clarity, e.g., “Seeds promote the rapid diversification of angiosperms by establishing postzygotic reproductive barriers, such as hybrid seed inviability.”
  • “Postzygotic barriers to reproduction are often less transient than prezygotic barriers”: This could be rephrased for greater clarity, e.g., "Postzygotic barriers tend to be more permanent compared to prezygotic barriers."

iii)           Terminology:

  • “pivotal role in ensuring speciation”: Consider rephrasing "ensuring speciation" to "facilitating speciation" for a more accurate description of the role of these barriers.
  • “gene regulatory network (GRN) inference”: The term "inference" might confuse some readers. Consider using "analysis" or "construction" for clarity.

iv)              Grammar and Word Choice:

  • “We deployed two GRN inference algorithms – RTP-STAR and KBoost –on three different subsets”: There is a missing space before "on." It should read "…algorithms – RTP-STAR and KBoost – on three different subsets."
  • “While the two algorithms yielded different results”: This could be more specific about what those results were to enhance the context.

v)             Repetition:

  • “identify active regulatory mechanisms” and “identify active regulatory relationships”: The use of "identify" in two consecutive sentences could be varied for better flow, perhaps using "uncover" or "determine" in one of the sentences.

vi)            Conclusion:

  • The last sentence could be more definitive about the implications of the findings. Instead of "potential novel regulatory mechanisms of interest for further investigation," it might be stronger to say "novel regulatory mechanisms that warrant further investigation."

b)    Introduction

i)                Line 19-20: "The evolutionary innovation of the seed as a reproductive propagule in seed plants is believed to have provided plants dramatic evolutionary advantages..."

  • The phrase "is believed to have provided" introduces uncertainty. If evidence supports this claim, consider using "has provided" for a stronger statement.
  • The word "dramatic" is vague and subjective. More specific terminology, such as "significant" or "substantial," would be clearer.

ii)             Line 21: "Some such conferred advantages specific to angiosperm seeds potentially spurred the rapid diversification..."

  • The phrase "Some such conferred advantages" is awkward. It would be clearer to state, "Certain advantages of angiosperm seeds" to improve readability.

iii)           Line 26-27: "...additionally, it influences timing of germination through abscisic acid (ABA) signaling."

  • The word "additionally" could be replaced with "furthermore" for a smoother flow.
  • "Timing of germination" could be specified as "the timing of seed germination" for better clarity.

iv)            Line 29-31: "Additionally, since endosperm failure is a common cause of hybrid seed inviability... has likely been responsible or the formation of many reproductive barriers..."

  • The sentence starts with "Additionally" again, which is repetitive. Use alternatives like "Moreover" or "Furthermore."
  • There is a typo in "responsible or the formation" — it should be "responsible for the formation."

v)             Line 32: "...this suggests that the endosperm holds a central role in explaining the remarkable species richness..."

  • "Holds a central role in explaining" could be simplified to "plays a central role in."

vi)            Line 36-37: "...and may discover new causes of seed failure."

  • The word "discover" is not the best fit here. "Identify" would be more precise.

vii)          Line 44-45: "...where a first, source gene codes for a transcription factor that binds to the promoter of a second, target gene..."

  • The phrase "a first, source gene" is awkward. It would be clearer to use "a source gene."

viii)        Line 49-50: "...a source gene directly regulating a gene that encodes a transcription factor which, in turn, directly regulates the target gene."

  • The sentence structure is a bit convoluted. Simplifying to "a source gene regulating another gene encoding a transcription factor that subsequently regulates the target gene" could improve readability.

ix)            Line 53-54: "GRN inference has the capacity to uncover previously unknown regulatory interactions, and has been used to highlight and discover multiple key regulatory genes..."

  • "Highlight and discover" is redundant. Use "identify and highlight."

x)             Line 83: "...the evolution of the diversity of angiosperms, Darwin’s famous 'abominable mystery'."

  • The phrase "the evolution of the diversity" is awkward. It can be shortened to "the evolution and diversity."

c)     Materials and methods

The "Materials and Methods" section presented here has a few issues that could be improved for clarity, consistency, and accuracy. Here are some mistakes and suggestions for improvement:

  1. Ambiguous Terminology:
    • The use of terms like "endosperm enriched," "whole seed," and "unfertilized ovules" should be clearly defined. For instance, stating explicitly what "endosperm enriched" means in the context of this study could help improve clarity.
  2. Inconsistent Use of Abbreviations:
    • The abbreviation "DAP" (days after pollination) is introduced without initial expansion, which might confuse readers unfamiliar with the term. It would be better to use the full term at first, followed by the abbreviation in parentheses.
    • The abbreviation "GRN" (gene regulatory network) appears in the text before being introduced. It should be defined when mentioned for the first time.
  3. Inconsistent Data Descriptions:
    • The number of genes used for the differential expression analysis varies. For example, it is stated that 24,203 genes returned detectable expression levels, but later mentions refer to filtering from 24,302 genes. This inconsistency needs to be resolved.
    • Additionally, some gene sets are described vaguely, like "MSE filtered only." It should be clear how Modified Shannon Entropy filtering is applied in each case.
  4. Lack of Details in RNA-seq Processing:
    • More details could be provided about the RNA-seq data preprocessing, such as specific trimming parameters used in Trimmomatic or alignment settings in STAR. Without these, the reproducibility of the analysis could be limited.
  5. Inadequate Explanation of GRN Inference Methods:
    • The explanation of GRN inference methods, such as RTP-STAR and KBoost, is very brief and lacks details about parameter settings or specific steps taken during the analysis. More information should be provided on why certain iterations were used and the rationale behind method selection.
  6. Potential Inaccuracy in Differential Expression Analysis Thresholds:
    • The thresholds used for differential expression (log2 fold change > 1.5 and FDR < 0.01) may not align with standard practices or other parts of the study. It’s important to clarify why these specific thresholds were chosen and whether they were based on previous studies or established criteria.
  7. Unclear Filtering and Annotation Processes:
    • The description of entropy filtering and transcription factor annotation is not sufficiently detailed. For example, how were outlier genes retained? What specific criteria were used in the annotation process for transcription factors? Providing step-by-step explanations would improve transparency.
  8. Omission of Quality Control Details:
    • While the use of tools like FastQC is mentioned, there is no discussion of the actual quality control metrics obtained or actions taken based on them (e.g., percentage of trimmed reads). Including such details would strengthen the rigor of the methods.
  9. Overuse of Acronyms Without Explanation:
    • The acronyms like GO, PFam, and FPKM are used without expanding them or providing context for readers unfamiliar with the terms. A brief description of what they stand for and their significance in the study would be helpful.
  10. Network Motif Analysis Could Be Expanded:
  • The description of the network motif analysis with QuateXelero is brief and does not explain how motif significance was determined or what types of motifs were found. Including more details about this analysis could provide deeper insights into the GRN results.

d)    Results

There are several issues within the results section that should be addressed to improve clarity and accuracy:

  1. Inconsistent Terminology:
    • The terms "RNAseq samplest" and "RNAseq samplest" appear to be typographical errors. The correct term is "RNA-seq samples."
  2. Inconsistent Formatting:
    • There is inconsistent spacing in some places, such as "RNAseq samplest and" instead of "RNA-seq samples and." This could confuse readers.
    • The text sometimes uses abbreviations (like RNAseq) without hyphenation, while other instances use hyphenation (RNA-seq). It's best to use one format consistently.
  3. Typographical Errors:
    • The phrase "and subsequently filtered using a modified Shannon entropy approach" needs clarification. A more accurate phrasing might be, "and were subsequently filtered using a modified Shannon entropy approach."
    • "Inference Algorithhms" is a typo and should be corrected to "Algorithms."
  4. Unclear Gene Set Descriptions:
    • In describing the gene sets, it's important to clarify which gene set was filtered with which criteria and for what purpose. For instance, stating "subsequently filtered using a modified Shannon entropy approach (used to infer GRN1 and GRN2)" may confuse readers without further context on why or how these sets were chosen.
    • The phrase "comprised of all genes annotated as transcription factors" would be clearer if rephrased as "consisting of all genes annotated as transcription factors."
  5. Results Lacking Detail:
    • The statement, "For each inferred GRN we detected 3-node network motifs that were overrepresented in the network," lacks detail on what these motifs represent or imply biologically. More context about the biological significance of these motifs would strengthen the section.
    • The phrase, "14% were categorized as endosperm enriched through a differential expression analysis," should specify the total number of genes being considered for this percentage.
  6. Ambiguous References to Figures and Supplementary Files:
    • References to figures such as "1 and Methods," "Supplementary figures 1-6," or "3" are unclear. It's better to specify which figures support the results discussed, or provide specific legends for the referenced figures.
    • Mentioning supplementary files should be more informative, indicating what type of information is contained there.
  7. Vague Statistical Testing Description:
    • The phrase, "Permutation testing was performed for each pair of inferred networks" lacks detail about how many permutations were done or how the results are interpreted. Adding specifics about the permutation testing approach would clarify this section.
  8. Inconsistent Reference to Genes:
    • When discussing gene homologs (e.g., "HAIKU1 homolog MgTOL.E0874"), it would help to provide more context or references regarding the relevance of these homologs in the model organism or other species.
  9. Unclear Findings on Network Motif Detection:
    • The results state, "Network motifs were determined to be overrepresented... if they had a Z-score over 3.0." However, this could be more informative by explaining the biological implications or interpretations of these overrepresented motifs.
  10. Complex Phrasing:
  • The sentence, "Despite these systematic differences that impacted GRN topology, there were common regulations inferred using both RTP-STAR and KBoost when working with the same gene set," could be simplified to: "Both RTP-STAR and KBoost inferred similar regulations when using the same gene set, despite systematic differences affecting GRN topology."

e)     Discussion

-       Line 243-244: "While many of these putative regulations could spur further investigation, one specific motif of interest identified in GRN1 involves a feedback loop between two genes that is regulated by a third mediator gene (depicted in Table 1 motif 2)" – This sentence is a bit convoluted. The use of "putative regulations could spur further investigation" is vague. It would be clearer to specify what kind of investigations could be spurred and explicitly state the significance of the motif.

-       Line 254-256: "Therefore, both MgTOL.F0024 and MgTOL.E0523 are homologous to genes linked to the GA signaling pathway in A. thaliana, and are specifically homologous to genes lying on opposite ends of that cascade (FUS3 acting upstream of GA biosynthesis, and ZFP4 acting downstream of GA." – The sentence ends abruptly without proper closure, possibly due to a missing closing parenthesis or additional explanation. The phrase "lying on opposite ends of that cascade" is also ambiguous and should be clarified.

-       Line 257: "MgTOL.F0024 and MgTOL.E0523 may be involved in GA signaling and regulation related processes during endosperm and or seed development in Mimulus species." – The phrase "and or" is redundant; it should be "and/or" or just "or."

-       Line 259: "A caveat is that inferred regulations predicted in these GRNs may represent indirect relationships between genes." – This is presented as an afterthought. The statement could be integrated earlier in the section to better manage expectations regarding the limitations of inferred regulations.

-       Line 271: "signaling in the developing Arabidopsis endosperm [45]." – The mention of Arabidopsis could be clearer if it were explicitly stated how it relates to Mimulus or how findings from Arabidopsis can be used in this study.

-       Line 274-275: "allowed us to circumvent many – though not all – of the thresholding issues" – This phrasing is somewhat informal for an academic paper. A more precise term could be used instead of "circumvent," such as "overcome" or "address."

-       Line 277-278: "Though the connection is likely indirect, expression of PEG2 and its impact on hybrid seed failure seem to be linked to auxin production" – There is ambiguity in this sentence, especially with the phrase "connection is likely indirect." It would be beneficial to clarify the nature of the indirect relationship.

-       Line 301-302: "species are different. ." – There are two punctuation marks at the end. One period should be removed.

-       Line 303-304: "Given the interval between the developmental time series (approximately two days) it is likely that the regulatory relationships predicted in these GRNs include both direct and indirect relationships" – This sentence lacks a comma after "two days," which affects readability.

-       Line 306-307: "However, the biological importance of indirect relationships or relationships that may require the action of additional gene products between the identified node genes may still be significant and useful in the process of defining modules of important functional network motifs." – The sentence is too long and complex. It could be split into two sentences for better clarity.

Conclusion: No conclusion

Comments on the Quality of English Language

Title: “Machine Learning Inference of Gene Regulatory Networks in Developing Mimulus Seeds” correct to “Machine Learning Inference of Gene Regulatory Networks During Mimulus Seed Development”

a)    Abstract

i)               Clarity and Specificity:

  • “The angiosperm seed represents a critical evolutionary breakthrough”: It could be clearer to specify what this breakthrough entails, such as explaining how it facilitates reproductive success.
  • “likely helped propel the reproductive success and radiation of flowering plants”: The phrase "likely helped" introduces uncertainty. If there is strong evidence, it would be more assertive to say "has been shown to propel."

ii)             Sentence Structure:

  • “One way seeds facilitate the rapid diversification of angiosperms is through establishing postzygotic reproductive barriers in the form of hybrid seed inviability”: This sentence is somewhat convoluted. It could be simplified for clarity, e.g., “Seeds promote the rapid diversification of angiosperms by establishing postzygotic reproductive barriers, such as hybrid seed inviability.”
  • “Postzygotic barriers to reproduction are often less transient than prezygotic barriers”: This could be rephrased for greater clarity, e.g., "Postzygotic barriers tend to be more permanent compared to prezygotic barriers."

iii)           Terminology:

  • “pivotal role in ensuring speciation”: Consider rephrasing "ensuring speciation" to "facilitating speciation" for a more accurate description of the role of these barriers.
  • “gene regulatory network (GRN) inference”: The term "inference" might confuse some readers. Consider using "analysis" or "construction" for clarity.

iv)              Grammar and Word Choice:

  • “We deployed two GRN inference algorithms – RTP-STAR and KBoost –on three different subsets”: There is a missing space before "on." It should read "…algorithms – RTP-STAR and KBoost – on three different subsets."
  • “While the two algorithms yielded different results”: This could be more specific about what those results were to enhance the context.

v)             Repetition:

  • “identify active regulatory mechanisms” and “identify active regulatory relationships”: The use of "identify" in two consecutive sentences could be varied for better flow, perhaps using "uncover" or "determine" in one of the sentences.

vi)            Conclusion:

  • The last sentence could be more definitive about the implications of the findings. Instead of "potential novel regulatory mechanisms of interest for further investigation," it might be stronger to say "novel regulatory mechanisms that warrant further investigation."

b)    Introduction

i)                Line 19-20: "The evolutionary innovation of the seed as a reproductive propagule in seed plants is believed to have provided plants dramatic evolutionary advantages..."

  • The phrase "is believed to have provided" introduces uncertainty. If evidence supports this claim, consider using "has provided" for a stronger statement.
  • The word "dramatic" is vague and subjective. More specific terminology, such as "significant" or "substantial," would be clearer.

ii)             Line 21: "Some such conferred advantages specific to angiosperm seeds potentially spurred the rapid diversification..."

  • The phrase "Some such conferred advantages" is awkward. It would be clearer to state, "Certain advantages of angiosperm seeds" to improve readability.

iii)           Line 26-27: "...additionally, it influences timing of germination through abscisic acid (ABA) signaling."

  • The word "additionally" could be replaced with "furthermore" for a smoother flow.
  • "Timing of germination" could be specified as "the timing of seed germination" for better clarity.

iv)            Line 29-31: "Additionally, since endosperm failure is a common cause of hybrid seed inviability... has likely been responsible or the formation of many reproductive barriers..."

  • The sentence starts with "Additionally" again, which is repetitive. Use alternatives like "Moreover" or "Furthermore."
  • There is a typo in "responsible or the formation" — it should be "responsible for the formation."

v)             Line 32: "...this suggests that the endosperm holds a central role in explaining the remarkable species richness..."

  • "Holds a central role in explaining" could be simplified to "plays a central role in."

vi)            Line 36-37: "...and may discover new causes of seed failure."

  • The word "discover" is not the best fit here. "Identify" would be more precise.

vii)          Line 44-45: "...where a first, source gene codes for a transcription factor that binds to the promoter of a second, target gene..."

  • The phrase "a first, source gene" is awkward. It would be clearer to use "a source gene."

viii)        Line 49-50: "...a source gene directly regulating a gene that encodes a transcription factor which, in turn, directly regulates the target gene."

  • The sentence structure is a bit convoluted. Simplifying to "a source gene regulating another gene encoding a transcription factor that subsequently regulates the target gene" could improve readability.

ix)            Line 53-54: "GRN inference has the capacity to uncover previously unknown regulatory interactions, and has been used to highlight and discover multiple key regulatory genes..."

  • "Highlight and discover" is redundant. Use "identify and highlight."

x)             Line 83: "...the evolution of the diversity of angiosperms, Darwin’s famous 'abominable mystery'."

  • The phrase "the evolution of the diversity" is awkward. It can be shortened to "the evolution and diversity."

c)     Materials and methods

The "Materials and Methods" section presented here has a few issues that could be improved for clarity, consistency, and accuracy. Here are some mistakes and suggestions for improvement:

  1. Ambiguous Terminology:
    • The use of terms like "endosperm enriched," "whole seed," and "unfertilized ovules" should be clearly defined. For instance, stating explicitly what "endosperm enriched" means in the context of this study could help improve clarity.
  2. Inconsistent Use of Abbreviations:
    • The abbreviation "DAP" (days after pollination) is introduced without initial expansion, which might confuse readers unfamiliar with the term. It would be better to use the full term at first, followed by the abbreviation in parentheses.
    • The abbreviation "GRN" (gene regulatory network) appears in the text before being introduced. It should be defined when mentioned for the first time.
  3. Inconsistent Data Descriptions:
    • The number of genes used for the differential expression analysis varies. For example, it is stated that 24,203 genes returned detectable expression levels, but later mentions refer to filtering from 24,302 genes. This inconsistency needs to be resolved.
    • Additionally, some gene sets are described vaguely, like "MSE filtered only." It should be clear how Modified Shannon Entropy filtering is applied in each case.
  4. Lack of Details in RNA-seq Processing:
    • More details could be provided about the RNA-seq data preprocessing, such as specific trimming parameters used in Trimmomatic or alignment settings in STAR. Without these, the reproducibility of the analysis could be limited.
  5. Inadequate Explanation of GRN Inference Methods:
    • The explanation of GRN inference methods, such as RTP-STAR and KBoost, is very brief and lacks details about parameter settings or specific steps taken during the analysis. More information should be provided on why certain iterations were used and the rationale behind method selection.
  6. Potential Inaccuracy in Differential Expression Analysis Thresholds:
    • The thresholds used for differential expression (log2 fold change > 1.5 and FDR < 0.01) may not align with standard practices or other parts of the study. It’s important to clarify why these specific thresholds were chosen and whether they were based on previous studies or established criteria.
  7. Unclear Filtering and Annotation Processes:
    • The description of entropy filtering and transcription factor annotation is not sufficiently detailed. For example, how were outlier genes retained? What specific criteria were used in the annotation process for transcription factors? Providing step-by-step explanations would improve transparency.
  8. Omission of Quality Control Details:
    • While the use of tools like FastQC is mentioned, there is no discussion of the actual quality control metrics obtained or actions taken based on them (e.g., percentage of trimmed reads). Including such details would strengthen the rigor of the methods.
  9. Overuse of Acronyms Without Explanation:
    • The acronyms like GO, PFam, and FPKM are used without expanding them or providing context for readers unfamiliar with the terms. A brief description of what they stand for and their significance in the study would be helpful.
  10. Network Motif Analysis Could Be Expanded:
  • The description of the network motif analysis with QuateXelero is brief and does not explain how motif significance was determined or what types of motifs were found. Including more details about this analysis could provide deeper insights into the GRN results.

d)    Results

There are several issues within the results section that should be addressed to improve clarity and accuracy:

  1. Inconsistent Terminology:
    • The terms "RNAseq samplest" and "RNAseq samplest" appear to be typographical errors. The correct term is "RNA-seq samples."
  2. Inconsistent Formatting:
    • There is inconsistent spacing in some places, such as "RNAseq samplest and" instead of "RNA-seq samples and." This could confuse readers.
    • The text sometimes uses abbreviations (like RNAseq) without hyphenation, while other instances use hyphenation (RNA-seq). It's best to use one format consistently.
  3. Typographical Errors:
    • The phrase "and subsequently filtered using a modified Shannon entropy approach" needs clarification. A more accurate phrasing might be, "and were subsequently filtered using a modified Shannon entropy approach."
    • "Inference Algorithhms" is a typo and should be corrected to "Algorithms."
  4. Unclear Gene Set Descriptions:
    • In describing the gene sets, it's important to clarify which gene set was filtered with which criteria and for what purpose. For instance, stating "subsequently filtered using a modified Shannon entropy approach (used to infer GRN1 and GRN2)" may confuse readers without further context on why or how these sets were chosen.
    • The phrase "comprised of all genes annotated as transcription factors" would be clearer if rephrased as "consisting of all genes annotated as transcription factors."
  5. Results Lacking Detail:
    • The statement, "For each inferred GRN we detected 3-node network motifs that were overrepresented in the network," lacks detail on what these motifs represent or imply biologically. More context about the biological significance of these motifs would strengthen the section.
    • The phrase, "14% were categorized as endosperm enriched through a differential expression analysis," should specify the total number of genes being considered for this percentage.
  6. Ambiguous References to Figures and Supplementary Files:
    • References to figures such as "1 and Methods," "Supplementary figures 1-6," or "3" are unclear. It's better to specify which figures support the results discussed, or provide specific legends for the referenced figures.
    • Mentioning supplementary files should be more informative, indicating what type of information is contained there.
  7. Vague Statistical Testing Description:
    • The phrase, "Permutation testing was performed for each pair of inferred networks" lacks detail about how many permutations were done or how the results are interpreted. Adding specifics about the permutation testing approach would clarify this section.
  8. Inconsistent Reference to Genes:
    • When discussing gene homologs (e.g., "HAIKU1 homolog MgTOL.E0874"), it would help to provide more context or references regarding the relevance of these homologs in the model organism or other species.
  9. Unclear Findings on Network Motif Detection:
    • The results state, "Network motifs were determined to be overrepresented... if they had a Z-score over 3.0." However, this could be more informative by explaining the biological implications or interpretations of these overrepresented motifs.
  10. Complex Phrasing:
  • The sentence, "Despite these systematic differences that impacted GRN topology, there were common regulations inferred using both RTP-STAR and KBoost when working with the same gene set," could be simplified to: "Both RTP-STAR and KBoost inferred similar regulations when using the same gene set, despite systematic differences affecting GRN topology."

e)     Discussion

-       Line 243-244: "While many of these putative regulations could spur further investigation, one specific motif of interest identified in GRN1 involves a feedback loop between two genes that is regulated by a third mediator gene (depicted in Table 1 motif 2)" – This sentence is a bit convoluted. The use of "putative regulations could spur further investigation" is vague. It would be clearer to specify what kind of investigations could be spurred and explicitly state the significance of the motif.

-       Line 254-256: "Therefore, both MgTOL.F0024 and MgTOL.E0523 are homologous to genes linked to the GA signaling pathway in A. thaliana, and are specifically homologous to genes lying on opposite ends of that cascade (FUS3 acting upstream of GA biosynthesis, and ZFP4 acting downstream of GA." – The sentence ends abruptly without proper closure, possibly due to a missing closing parenthesis or additional explanation. The phrase "lying on opposite ends of that cascade" is also ambiguous and should be clarified.

-       Line 257: "MgTOL.F0024 and MgTOL.E0523 may be involved in GA signaling and regulation related processes during endosperm and or seed development in Mimulus species." – The phrase "and or" is redundant; it should be "and/or" or just "or."

-       Line 259: "A caveat is that inferred regulations predicted in these GRNs may represent indirect relationships between genes." – This is presented as an afterthought. The statement could be integrated earlier in the section to better manage expectations regarding the limitations of inferred regulations.

-       Line 271: "signaling in the developing Arabidopsis endosperm [45]." – The mention of Arabidopsis could be clearer if it were explicitly stated how it relates to Mimulus or how findings from Arabidopsis can be used in this study.

-       Line 274-275: "allowed us to circumvent many – though not all – of the thresholding issues" – This phrasing is somewhat informal for an academic paper. A more precise term could be used instead of "circumvent," such as "overcome" or "address."

-       Line 277-278: "Though the connection is likely indirect, expression of PEG2 and its impact on hybrid seed failure seem to be linked to auxin production" – There is ambiguity in this sentence, especially with the phrase "connection is likely indirect." It would be beneficial to clarify the nature of the indirect relationship.

-       Line 301-302: "species are different. ." – There are two punctuation marks at the end. One period should be removed.

-       Line 303-304: "Given the interval between the developmental time series (approximately two days) it is likely that the regulatory relationships predicted in these GRNs include both direct and indirect relationships" – This sentence lacks a comma after "two days," which affects readability.

-       Line 306-307: "However, the biological importance of indirect relationships or relationships that may require the action of additional gene products between the identified node genes may still be significant and useful in the process of defining modules of important functional network motifs." – The sentence is too long and complex. It could be split into two sentences for better clarity.

Conclusion: No conclusion

Author Response

Abstract

Comments 1:  Clarity and Specificity of “The angiosperm seed represents a critical evolutionary breakthrough”: It could be clearer to specify what this breakthrough entails, such as explaining how it facilitates reproductive success.

Revisions 1: We agreed with and made the suggested revision from "likely helped" to "has been shown" (page 1, line 1).

Comments 2: “One way seeds facilitate the rapid diversification of angiosperms is through establishing postzygotic reproductive barriers in the form of hybrid seed inviability”: This sentence is somewhat convoluted. It could be simplified for clarity, e.g., “Seeds promote the rapid diversification of angiosperms by establishing postzygotic reproductive barriers, such as hybrid seed inviability.”

Revisions 2: We agreed with and made the suggestions (page 1, lines 2-3)

Comments 3: “Postzygotic barriers to reproduction are often less transient than prezygotic barriers”: This could be rephrased for greater clarity, e.g., "Postzygotic barriers tend to be more permanent compared to prezygotic barriers."

Revisions 3: We agreed with and made the suggestions (page 1, lines 4-5)

Comments 4: “pivotal role in ensuring speciation”: Consider rephrasing "ensuring speciation" to "facilitating speciation" for a more accurate description of the role of these barriers.

Revisions 4: We agreed with and made the suggestions (page 1, lines 5)

Comments 5: “gene regulatory network (GRN) inference”: The term "inference" might confuse some readers. Consider using "analysis" or "construction" for clarity.

Revisions 5: We mostly agreed with the suggestions and changed it to "inference analysis" since gene regulatory network inference is the complete name of the type of analysis (page 1, lines 8).

Comments 6: “We deployed two GRN inference algorithms – RTP-STAR and KBoost –on three different subsets”: There is a missing space before "on." It should read "…algorithms – RTP-STAR and KBoost – on three different subsets."

Revisions 6: We agreed with and made the suggestions (page 1, lines 13)

Comments 7: “While the two algorithms yielded different results”: This could be more specific about what those results were to enhance the context.

Revisions 7: We agreed with the suggestions and changed to "While the two algorithms yielded GRNs with different regulations and topologies when working with the same data subset," (page 1, lines 14-15)

Comments 8: “identify active regulatory mechanisms” and “identify active regulatory relationships”: The use of "identify" in two consecutive sentences could be varied for better flow, perhaps using "uncover" or "determine" in one of the sentences.

Revisions 8: We agreed with the suggestions and changed to "uncover" (page 1, line 11)

Comments 9: The last sentence could be more definitive about the implications of the findings. Instead of "potential novel regulatory mechanisms of interest for further investigation," it might be stronger to say "novel regulatory mechanisms that warrant further investigation."

Revisions 9: We agreed with and made the suggestions (page 1, lines 16)

Comments 10: The last sentence could be more definitive about the implications of the findings. Instead of "potential novel regulatory mechanisms of interest for further investigation," it might be stronger to say "novel regulatory mechanisms that warrant further investigation."

Revisions 10: We agreed with and made the suggestions (page 1, lines 16)

Introduction

Comments 1: The phrase "is believed to have provided" introduces uncertainty. If evidence supports this claim, consider using "has provided" for a stronger statement.

Revisions 1: We agreed with and made the the suggested revisions (page 1, line 21).

Comments 2: The word "dramatic" is vague and subjective. More specific terminology, such as "significant" or "substantial," would be clearer.

Revisions 2: We agreed with and made the the suggested revision to "substantial" (page 1, line 21).

Coments 3: The phrase "Some such conferred advantages" is awkward. It would be clearer to state, "Certain advantages of angiosperm seeds" to improve readability.

Revisions 3: we agreed with and made the suggested revisions (page 1, line 22)

Comments 4: The word "additionally" could be replaced with "furthermore" for a smoother flow.

Revisions 4: we agreed with and made the suggested revisions (page 1, line 27).

Comments 5: "Timing of germination" could be specified as "the timing of seed germination" for better clarity.

Revisions 5: we agreed with and made the suggested revisions (page 1, line 27).

Comments 6: The sentence starts with "Additionally" again, which is repetitive. Use alternatives like "Moreover" or "Furthermore."

Revisions 6: we agreed with and made the suggested revision to "Moreover" (page 1, line 30).

Comments 7: There is a typo in "responsible or the formation" — it should be "responsible for the formation."

Revisions 7: we agreed with and made the suggested revisions (page 1, line 31).

Comments 8: "Holds a central role in explaining" could be simplified to "plays a central role in."

Revisions 8: we agreed with and made the suggested revisions (page 1, line 33).

Comments 9: "Holds a central role in explaining" could be simplified to "plays a central role in."

Revisions 9: we agreed with and made the suggested revisions (page 1, line 33).

Comments 10: The word "discover" is not the best fit here. "Identify" would be more precise.

Revisions 10: we agreed with and made the suggested revisions (page 1, line 38).

Comments 11: The phrase "a first, source gene" is awkward. It would be clearer to use "a source gene."

Revisions 11: we agreed with and made the suggested revisions (page 2, line 45).

Comments 12: The sentence structure is a bit convoluted. Simplifying to "a source gene regulating another gene encoding a transcription factor that subsequently regulates the target gene" could improve readability.

Revisions 12: we agreed with and made the suggested revisions (page 2, line 51).

Comments 13: "Highlight and discover" is redundant. Use "identify and highlight."

Revisions 13: we agreed with and made the suggested revisions (page 2, line 55).

Comments 14: The phrase "the evolution of the diversity" is awkward. It can be shortened to "the evolution and diversity."

Revisions 14: we agreed with and made the suggested revisions (page 2, line 84).

Methods

Comments 1: The use of terms like "endosperm enriched," "whole seed," and "unfertilized ovules" should be clearly defined. For instance, stating explicitly what "endosperm enriched" means in the context of this study could help improve clarity.

Revisions 1: We appreciate this consideration and went back to ensure that when terms like "whole seed" and "isolated endosperm" are first introduced in quotations as a descriptor to a set of genes, they are accompanied with brief explanations for clarity.

Comments 2: The abbreviation "DAP" (days after pollination) is introduced without initial expansion, which might confuse readers unfamiliar with the term. It would be better to use the full term at first, followed by the abbreviation in parentheses. The abbreviation "GRN" (gene regulatory network) appears in the text before being introduced. It should be defined when mentioned for the first time.

Revisions 2: We noted not only these instances but others such as FPKM where abbreviations needed to be introduced upon first use. We identified and revised the manuscript accordingly. Thank you for bringing this to our attention.

Comments 3: The number of genes used for the differential expression analysis varies. For example, it is stated that 24,203 genes returned detectable expression levels, but later mentions refer to filtering from 24,302 genes. This inconsistency needs to be resolved. Additionally, some gene sets are described vaguely, like "MSE filtered only." It should be clear how Modified Shannon Entropy filtering is applied in each case.

Revisions 3: Thank you for pointing out the typo in the gene count. The correct value is 24,203 and we revised the manuscript accordingly. Also, we agree "MSE filtered only" was unclear and changed it to "whole gene set with MSE filtering".

Comments 4: More details could be provided about the RNA-seq data preprocessing, such as specific trimming parameters used in Trimmomatic or alignment settings in STAR. Without these, the reproducibility of the analysis could be limited.

Revisions 4: We agree that settings and parameters used when deploying Trimmomatic and STAR are important to reproducibility, we added the settings used in parentheticals to each time they are referenced in the manuscript.

Comments 5: More details could be provided about the RNA-seq data preprocessing, such as specific trimming parameters used in Trimmomatic or alignment settings in STAR. Without these, the reproducibility of the analysis could be limited.

Revisions 5: We agree that settings and parameters used when deploying Trimmomatic and STAR are important to reproducibility, we added the settings used in parentheticals to each time they are referenced in the manuscript.

Comments 6: The explanation of GRN inference methods, such as RTP-STAR and KBoost, is very brief and lacks details about parameter settings or specific steps taken during the analysis. More information should be provided on why certain iterations were used and the rationale behind method selection.

Revisions 6: For both RTP-STAR deployed using tuxnet and KBoost, we did include details to each dataset we provided to the algorithm and what each dataset was used for (specifically the overall FPKM expression values calculated from STAR/Cufflinks used for main coexpression matrix for each algorithm, and then the average FPKM for each gene at each time point in the time-course was used as a time course component for RTP-STAR). The decision to switch from 5 iterations to 1 iteration of RTP-STAR was due to the fact that it becomes prohibitively time consuming to perform these inferences, and the use of multiple iterations is only recomended when spatio-temporal data is used for clustering (which we did not do). KBoost was deployed with default settings and no prior weightings applied, this is a point we further clarified in revisions.

Comments 7: The thresholds used for differential expression (log2 fold change > 1.5 and FDR < 0.01) may not align with standard practices or other parts of the study. It’s important to clarify why these specific thresholds were chosen and whether they were based on previous studies or established criteria.

Revisions 7: Thank you for drawing our attention to the discrepancy in the reported FDR thresholds used. This was errantly labeled as 0.05 in the figure legend for FIgure 2, when it should have been 0.01, in agreement with the FDR threshold value used in the rest of the paper and a commonly used threshold for differential expression analyses.

Comments 8: The description of entropy filtering and transcription factor annotation is not sufficiently detailed. For example, how were outlier genes retained? What specific criteria were used in the annotation process for transcription factors? Providing step-by-step explanations would improve transparency.

Revisions 8: Thank you for drawing our attention to this. The MSE scoring, outlier detection, and filtering steps were based upon previous work done by Thomas (2022) and we provided more detail as to how the steps were taken in our analysis on page 5 line 164.

Comments 9: While the use of tools like FastQC is mentioned, there is no discussion of the actual quality control metrics obtained or actions taken based on them (e.g., percentage of trimmed reads). Including such details would strengthen the rigor of the methods.

Revisions 9: FastQC was used as a check to ensure no reads flagged as low quality reads may have remained in the dataset after deploying Trimmomatic, we made sure to clarify this on page 3 line 118.

Comments 10: The acronyms like GO, PFam, and FPKM are used without expanding them or providing context for readers unfamiliar with the terms. A brief description of what they stand for and their significance in the study would be helpful.

Revisions 10: As mentioned in the response to Methods Comments 2. For these acronyms, we made sure to expand where applicable. For some of these such as PFam and InterPro, they represent names of databases of protein and gene sequence annotations, and are not acronyms.

Comments 11: The description of the network motif analysis with QuateXelero is brief and does not explain how motif significance was determined or what types of motifs were found. Including more details about this analysis could provide deeper insights into the GRN results.

Revisions 11: We agreed with this assessment and further expanded on our use of QuateXelero on page 6 lines 197-203.

Results

Comments 1: The terms "RNAseq samplest" and "RNAseq samplest" appear to be typographical errors. The correct term is "RNA-seq samples."

Revisions 1: We agreed with and made the suggestions (page 7, lines 224)

Comments 2: There is inconsistent spacing in some places, such as "RNAseq samplest and" instead of "RNA-seq samples and." This could confuse readers.The text sometimes uses abbreviations (like RNAseq) without hyphenation, while other instances use hyphenation (RNA-seq). It's best to use one format consistently.

Revisions 2: We agreed with and made the suggestions (page 7, lines 224) and where applicable for RNA-seq

Comments 3: The phrase "and subsequently filtered using a modified Shannon entropy approach" needs clarification. A more accurate phrasing might be, "and were subsequently filtered using a modified Shannon entropy approach." "Inference Algorithhms" is a typo and should be corrected to "Algorithms."

Revisions 3: We agreed with and made the suggestions (page 6, lines 213, and page 8 line 248).

Comments 4:In describing the gene sets, it's important to clarify which gene set was filtered with which criteria and for what purpose. For instance, stating "subsequently filtered using a modified Shannon entropy approach (used to infer GRN1 and GRN2)" may confuse readers without further context on why or how these sets were chosen. The phrase "comprised of all genes annotated as transcription factors" would be clearer if rephrased as "consisting of all genes annotated as transcription factors."

Revisions 4: We agreed with the suggestions and made the first changes on page 6 213, and changed the latter sentence to "consisted of all genes annotated as transcription factors" (page 6, lines 217)

Comments 5: The statement, "For each inferred GRN we detected 3-node network motifs that were overrepresented in the network," lacks detail on what these motifs represent or imply biologically. More context about the biological significance of these motifs would strengthen the section.The phrase, "14% were categorized as endosperm enriched through a differential expression analysis," should specify the total number of genes being considered for this percentage.

Revisions 5: These motifs were selected for their overrepresented nature in the motifs as an unbiased way of highlighting genes and regulations in the inferred GRNs. The use of this approach to uncover biologically important regulatory mechanisms and genetic relationships is presented in the introduction. We added the context of the total gene set size for the percentages

Comments 6: References to figures such as "1 and Methods," "Supplementary figures 1-6," or "3" are unclear. It's better to specify which figures support the results discussed, or provide specific legends for the referenced figures. Mentioning supplementary files should be more informative, indicating what type of information is contained there.

Revisions 6: We agreed with the suggestion and added more detail and context where applicable to places where figures or supplementary materials are referenced.

Comments 7: The phrase, "Permutation testing was performed for each pair of inferred networks" lacks detail about how many permutations were done or how the results are interpreted. Adding specifics about the permutation testing approach would clarify this section.

Revisions 7: We agreed with the suggestion and expanded upon how the permutation testing was performed and the specifics of how it was set up.

Comments 8: The phrase, "Permutation testing was performed for each pair of inferred networks" lacks detail about how many permutations were done or how the results are interpreted. Adding specifics about the permutation testing approach would clarify this section.

Revisions 8: We agreed with the suggestion and expanded upon how the permutation testing was performed and the specifics of how it was set up(page 9, lines 258-274).

Comments 9: The results state, "Network motifs were determined to be overrepresented... if they had a Z-score over 3.0." However, this could be more informative by explaining the biological implications or interpretations of these overrepresented motifs.

Revisions 9: The ability of enriched network motifs to identify regulatory mechanisms and genetic regulations of biological importance is described in the introduction, the decision to not pre-select known motifs was made to perform this in an unbiased manner. The process of the random network generation and motif detection with Quatexelero was expanded upon.

Comments 10: The sentence, "Despite these systematic differences that impacted GRN topology, there were common regulations inferred using both RTP-STAR and KBoost when working with the same gene set," could be simplified to: "Both RTP-STAR and KBoost inferred similar regulations when using the same gene set, despite systematic differences affecting GRN topology."

Revisions 10: We agreed with and made the changes according to the suggestion

Discussion

      Line 243-244: "While many of these putative regulations could spur further investigation, one specific motif of interest identified in GRN1 involves a feedback loop between two genes that is regulated by a third mediator gene (depicted in Table 1 motif 2)" – This sentence is a bit convoluted. The use of "putative regulations could spur further investigation" is vague. It would be clearer to specify what kind of investigations could be spurred and explicitly state the significance of the motif.

This was changed to "Many of these putative regulations could spur further investigation as potential active regulatory mechanisms in the developing Mimulus seed."

  •       Line 254-256: "Therefore, both MgTOL.F0024 and MgTOL.E0523 are homologous to genes linked to the GA signaling pathway in A. thaliana, and are specifically homologous to genes lying on opposite ends of that cascade (FUS3 acting upstream of GA biosynthesis, and ZFP4 acting downstream of GA." – The sentence ends abruptly without proper closure, possibly due to a missing closing parenthesis or additional explanation. The phrase "lying on opposite ends of that cascade" is also ambiguous and should be clarified.
  • This was changed to provide more clarity in the paranthetical it is now "Therefore, both MgTOL.F0024 and MgTOL.E0523 are homologous to genes linked to the GA signaling pathway in A. thaliana, and are specifically homologous to genes lying on opposite ends of that cascade (FUS3 acts upstream of GA biosynthesis, while ZFP4 acts downstream of GA)".
  •       Line 257: "MgTOL.F0024 and MgTOL.E0523 may be involved in GA signaling and regulation related processes during endosperm and or seed development in Mimulus species." – The phrase "and or" is redundant; it should be "and/or" or just "or."
  • This change was made to "or".
  •       Line 259: "A caveat is that inferred regulations predicted in these GRNs may represent indirect relationships between genes." – This is presented as an afterthought. The statement could be integrated earlier in the section to better manage expectations regarding the limitations of inferred regulations.
  • This was moved up to the beginning of the discussion at the end of the first paragraph.
  •       Line 271: "signaling in the developing Arabidopsis endosperm [45]." – The mention of Arabidopsis could be clearer if it were explicitly stated how it relates to Mimulus or how findings from Arabidopsis can be used in this study.
  • Arabidopsis is invoked as a commonly referenced comparison organism in angiosperm studies. It is referenced due to the abundance of information available on the organism rather than a specific phylogenetic relationship we are interested in highlighting.
  •       Line 274-275: "allowed us to circumvent many – though not all – of the thresholding issues" – This phrasing is somewhat informal for an academic paper. A more precise term could be used instead of "circumvent," such as "overcome" or "address."
  • This was changed to "overcome"
  •       Line 277-278: "Though the connection is likely indirect, expression of PEG2 and its impact on hybrid seed failure seem to be linked to auxin production" – There is ambiguity in this sentence, especially with the phrase "connection is likely indirect." It would be beneficial to clarify the nature of the indirect relationship.
  • Here, we feel that the connection being indirect reflects what the relationship is known to be. Further analysis is required before we can be more precise in describing this relationsip.
  •       Line 301-302: "species are different. ." – There are two punctuation marks at the end. One period should be removed.
  • This was changed to remove second period.
  •       Line 303-304: "Given the interval between the developmental time series (approximately two days) it is likely that the regulatory relationships predicted in these GRNs include both direct and indirect relationships" – This sentence lacks a comma after "two days," which affects readability.
  • This was changed to include comma
  •       Line 306-307: "However, the biological importance of indirect relationships or relationships that may require the action of additional gene products between the identified node genes may still be significant and useful in the process of defining modules of important functional network motifs." – The sentence is too long and complex. It could be split into two sentences for better clarity.
  • This was changed to "However, the biological importance of indirect relationships may still be significant and useful in the process of defining modules of important functional network motifs"

Reviewer 2 Report

Comments and Suggestions for Authors

In this manuscirpt, the authors ananlyzed the gene regulatory networks in developing Mimulus seeds by the RNA-seq data by Machine learning method. It is an interesting paper, and I sugget to publish with minor revisions.

In the Introduction, the autors should talk more about the Machine Learning, and different methods have been used in analyzing the RNA-seq data, and why chose RTP-ATAR adn KBoost.

And also in the discussion, the authors should discuss more compare to the previous paper 14, since the authors were using the RNA-seq data from that paper.

Author Response

Comments 1: In the Introduction, the autors should talk more about the Machine Learning, and different methods have been used in analyzing the RNA-seq data, and why chose RTP-STAR and KBoost.

Revisions 1: Thank you for pointing out that the machine learning components of our chosen algorithms warranted further discussion. We made sure to add more detail to the nature of the algorithms, and elaborate why they were chosen in the introduction (pages 2-3, lines 86-105). In short, RTP-STAR is an expansion of the GENIE3 algorithm, which was among the highest scoring GRN inference algorithms for the DREAM4 and DREAM5 GRN inference challenges. RTP-STAR builds upon GENIE3 by allowing for the incorporation of temporal expression data, which lent it self to the time-course nature of our dataset. On the other hand, KBoost was selected because it has demonstrated performance that is similar or better than the algorithms benchmarked in the DREAM4 and DREAM5 GRN inference challenges, while also demonstrating substantial improvements in computational efficiency. Since two of the three gene sets we were investigating were quite large, this made KBoost well fit for our analysis.

Comments 2: And also in the discussion, the authors should discuss more compare to the previous paper 14, since the authors were using the RNA-seq data from that paper.

Revisions 2: Thank you for pointing out that we could make more clear how this analysis has extended the findings of the previous paper. We have added a lead paragraph to the discussion elaborating on this (page 9, lines 272-281).

Reviewer 3 Report

Comments and Suggestions for Authors

In this study entitled “Machine Learning Inference of Gene Regulatory Networks in Developing Mimulus Seeds”, the authors performed GRN inference to explore the gene regulatory mechanisms in Mimulus seeds, and developed two GRN inference algorithms based on three different subsets of transcriptomic dataset. This manuscript is an original work and written well, and it is a very comfortable to read, their logic is very clear. This job provides valuable reference for studying regulatory networks in Mimulus. 

1.     Different results were found from RTP-STAR and KBoost algorithms, which algorithm is more feasible, and whether the results of these two algorithms overlap?

2.     The authors identified multiple regulatory genes in Mimulus seeds, and discussed their functions based on the homologs in A. thaliana. Whether these genes have been reported functions in Mimulus, if so, please add to increase the reliability in this study.

3.     “significantly” need a statistical test, otherwise, use “validly”, “importantly”, “availably”, “remarkably” or “obviously”. Please checked them carefully.

4.     In general, the HISAT2+stringTie+Ballgrown process is very good for transcriptome analysis. Do the authors consider replacing STAR+Cufflinks/ STAR+featureCounts with this process?

5.     The “p” should be italics for “p-value”, and using “log2 (fold change)” to replace “log2 fold change”. Please check the full-text to ensure proper writing.

Author Response

Comments 1: Different results were found from RTP-STAR and KBoost algorithms, which algorithm is more feasible, and whether the results of these two algorithms overlap?

Response 1: Thank you for pointing out the comparison and contrast between RTP-STAR and KBoost could be clearer. We made a point to more explicitly detail why RTP-STAR and KBoost were chosen in the introduction (pages 2-3, lines 86-105). And added further elaboration on the results on the parametric testing done to compare the level of overlap between the two algorithms (page 9, lines 254-270).  In short, RTP-STAR was selected as it is an extension of one of the best performing GRN inference algorithms, GENIE3, from the DREAM4 and DREAM5 GRN inference challenges. And RTP-STAR builds upon GENIE3 by allowing for incorporation of temporal information about the samples (which lended itself to being utilized for analyzing our time-course dataset). KBoost was the second algorithm selected as it performs favorably relative to other benchmarked GRN inference algorithms, while also being considerably more computationally efficient when working with large gene sets like we are here.

Comments 2: The authors identified multiple regulatory genes in Mimulus seeds, and discussed their functions based on the homologs in A. thaliana. Whether these genes have been reported functions in Mimulus, if so, please add to increase the reliability in this study.

Response 2: For many of our highlighted genes, we have identified known roles in Mimulus where we are confident. For example, MgTOL.N1182 the homolog of ZHOUPI in Arabidopsis, has been independently shown to have endosperm-specific expression. However, we do hope that the inability to confidently assess the roles of the Mimulus homologs we discussed here represents potential upside for our analysis. As it is highlighting potential active regulatory mechanisms with known homologs in other angiosperm species.

Comments 3: “significantly” need a statistical test, otherwise, use “validly”, “importantly”, “availably”, “remarkably” or “obviously”. Please checked them carefully.

Response 3: Thank you for pointing this out. We have ensured every use of the terms "significant" and "significantly" are referencing something backed by a statistical test.

Comments 4: In general, the HISAT2+stringTie+Ballgrown process is very good for transcriptome analysis. Do the authors consider replacing STAR+Cufflinks/ STAR+featureCounts with this process?

Response 4: We did consider a number of other RNA-seq read aligners and accompanying pipelines. We decided upon STAR and Cufflinks/featureCounts for two primary reasons: 1) for one of the algorithms used (RTP-STAR through  tuxnet) this pipeline is recommended for data pre-preparation, and 2) we were confident in the performance of STAR as an aligner specific for aligning RNA-seq reads relative to other popular aligners.

Comments 5: The “p” should be italics for “p-value”, and using “log2 (fold change)” to replace “log2 fold change”. Please check the full-text to ensure proper writing.

Response 5: Thank you for pointing this out, we have ensured every use of "p-value" has the "p" italicized, and that every use of log2 fold change is written as log2(fold change).

Round 2

Reviewer 3 Report

Comments and Suggestions for Authors

I am pleased that the authors have modified and responded my comments, these is no comment to the current revision